# Dipyridylmethane Ethers as Ligands for Luminescent Ir Complexes

**DOI:** 10.3390/molecules26237161

**Published:** 2021-11-26

**Authors:** Giorgio Volpi, Claudio Garino, Roberto Gobetto, Carlo Nervi

**Affiliations:** Department of Chemistry, University of Turin, Via Pietro Giuria 7, 10125 Torino, Italy; claudio.garino@unito.it (C.G.); roberto.gobetto@unito.it (R.G.); carlo.nervi@unito.it (C.N.)

**Keywords:** iridium, ether ligands, luminescence, heteroleptic cyclometalated complexes, quantum yield, lifetime

## Abstract

This work reports two new cationic heteroleptic cyclometalated iridium complexes, containing ether derivatives of di(pyridin-2-yl)methanol. The new ligands are based on dipyridin-2-ylmethane and are designed to obtain ether-based intermediates with extended electronic conjugation by insertion of π system such as phenyl, allyl and ethynyl. Different synthetic strategies were employed to introduce these units, as molecular wires, between the dipyridin-2-ylmethane chelating portion and the terminal N-containing functional group, such as amine and carbamide. The corresponding complexes show luminescence in the blue region of the spectrum, lifetimes between 0.6 and 2.1 μs, high quantum yield and good electrochemical behavior. The computational description (DFT) of the electronic structure highlights the key role of the conjugated π systems on optical and electrochemical properties of the final products.

## 1. Introduction

*N*,*N*-bidentate ligands for coordination complexes, such as 2,2′-bipyridines and 1,10-phenanthrolines, play a pivotal role in key areas such as photovoltaics, lighting, molecular sensors, biological probes, catalysis, molecular electronics and supramolecular chemistry [1,2,3,4,5]. This variety of applications stem from the facts that such ligands, by chelating various transition metals, mix their and metal frontier orbitals, resulting in electron delocalization and interesting new physical, electrochemical and optical properties. Thus, the design and development of synthetic methodologies to obtain new *N*,*N*-bidentate ligands is of paramount importance.

In general, *N*,*N*-bidentate ligands represent a desirable class of compounds in the search for structural diversity with suitable performance. In particular, extending the electronic conjugation by insertion of π systems has been regarded as the most effective method to design new dyes and molecular wires. Several conjugated π systems, such as phenylene, thienylene and ethynyl groups have been employed for tuning structural, optical and electrochemical behaviors in the corresponding complexes [6,7,8,9]. In this context, the dipyridin-2-yl-methane skeleton is well known for being an efficient chelating structure which guarantees good optical properties in the corresponding metal complexes. We have previously developed different synthetic strategies to modify the dipyridin-2-ylmethane skeleton starting from di-2-pyridylketone [10,11]. In particular, we reported different modification introducing amino, imino, nitrile, hydrazine and alcoholic functional groups, to increase the coordination stability and tune the optical properties. Moreover, the direct cyclization of di-2-pyridylketone (to form imidazo[1,5-*a*]pyridine ligands) [12,13,14,15,16,17,18] is widely reported to prepare emissive metal complexes [19,20,21,22,23,24,25]. Furthermore, surface functionalization with luminescent ligands and corresponding complexes is attracting a high interest, with impact in catalysis, molecular electronics, analytical detection or sensor technology [26,27]. These applications require the presence of appropriate functional groups, such as amine or carbamide [28,29,30], that must be inserted peripherally on the metal complex [31,32,33].

In this work, di-2-pyridylketone plays a key role in the modification of di(pyridin-2-yl) type ligands. As previously reported by us, the reduction of the carbonyl group of the di-2-pyridylketone significantly enhances the luminescent properties in the corresponding complexes, since the energetically accessible antibonding C=O orbital is no longer available for excited states quenching [10,11]. Hence, here we focus on the reduced derivative of di-2-pyridylketone: the di(pyridin-2-yl)methanol (**L0**). In particular, the formation of ether derivatives from dipyridin-2-ylmethanol, by Williamson reaction, is the principal target. The obtained ether-based intermediates were modified through cross coupling reactions to extend the spacers between the di(pyridin-2-yl) ligand moiety and the terminal functional group, by insertion of highly conjugated units such as phenyl, allyl and ethynyl. Concerning the functionalization, we herein explore different synthetic pathways to obtain new ligands bearing tethered primary amino (**A1**–**4**, Figure 1) and carbamide protecting (**B1**, Figure 1) terminal groups suitable for functionalization. Five new ligands were obtained and reacted with the chloro-bridged iridium(III) binuclear precursor [Ir(ppy)_2_Cl]_2_ (ppy = 2-phenylpyridine), affording two new luminescent cyclometalated iridium complexes (Figure 2).

## 2. Results and Discussion

Ideally, the design of functionalizable ligands should include: a coordination site (here di(pyridin-2-yl)), a stable linker (here oxy-methylene), a tunable spacer and a terminal group suitable for conjugation reactions (Figure 3). The presence of terminal substituents, such as amine and carbamide, must be far enough from the chelating site to avoid any interference during the complexation and preserve the possibility of chemical or electrochemical functionalization.

### 2.1. Syntheses

All the new ligands were prepared starting from di(pyridin-2-yl)methanol (**L0**), employing different synthetic approaches (reaction Figure 1 and Figure 2). We were able to introduce different spacers between the di(pyridin-2-yl) ligand moiety and the terminal functional group (amino or carbamate). In particular, in products **A1** and **B1** a propyl chain was introduced as spacer between the ether C–O bond and the N-containing functional groups. The comparison between the complexation reactions of **A1** and **B1** with the iridium(III) chloro-bridged dimer [Ir(ppy)_2_Cl]_2_ provides useful insights. In particular, the free amino terminal group, present in **A1**, reacts with the precursor [Ir(ppy)_2_Cl]_2_ giving different products involved in solution equilibria. On the contrary, the ligand **B1** does not show any side reactions, giving the single compound [Ir(ppy)_2_(**B1**)]^+^, isolated and characterized as a stable chloride salt. Nevertheless, BOC cleavage to deprotect the amino group of [Ir(ppy)_2_(**B1**)]^+^ (see Figure 1) leads to de-complexation of the di(pyridin-2-yl) ligand moiety, due to acidic conditions, with formation of **A1** and [Ir(ppy)_2_]^+^.

This observation led us to consider different spacers to increase the rigidity and the distance between the ether junction and the terminal amino group (**A2**–**A4**). The amino group was introduced on a terminal aromatic ring, avoiding the competition with the di(pyridin-2-yl) core during the complexation step. These spacers were inserted through the formation of an ether C–O bond on the di(pyridin-2-yl) core: methyl-4-(prop-1-en-1-yl)phenyl (**A2**), prop-1-yn-1-ylphenyl (**A3**) and 1,2-di-p-tolylethyne (**A4**). Starting from di(pyridin-2-yl)methanol (**L0**), the intermediates were obtained through the Williamson synthesis (Figure 2), generating the ether junction between the dipyridin-2-ylmethane and the spacer. Finally, **A2**–**A4** were synthesized through Sonogashira (**A3** and **A4**) and Heck (**A2**) cross coupling reactions (Figure 2).

Unfortunately, **A2** and **A3** was instable during the successive complexation reaction with [Ir(ppy)_2_Cl]_2_. Clear evidence of the cleavage of the ether bond present in the employed ligands was collected. After complexation, a large amount of [Ir(ppy)_2_(di(2-pyridyl)ketone)]^+^ was isolated in the reaction medium as side product. Despite we are not able to provide a mechanistic interpretation for such experimental evidence, actually the cleavage of the C–O ether bond by pincer iridium complexes has been previously reported in similar reaction conditions [34,35]. On the contrary, the ligand **A4** reacts with [Ir(ppy)_2_Cl]_2_ giving the stable corresponding complex [Ir(ppy)_2_(**A4**)]^+^. Indeed, the catalytic cleavage of the ether C–O bonds in these conditions is observed only in the presence of alkene or alkyne vicinal to the oxy-methylene portion (such as for **A2** and A**3**). [Ir(ppy)_2_(**A4**)]^+^ shows the dipyridin-2-ylmethane ligand portion followed by the oxy-methylene junction and the phenyl-ethyn-phenyl spacer, as a molecular wire, and a terminal free amino group. In this case, the terminal amino group is sufficiently far from the metal center to avoid their interaction during the complexation step, avoiding competing complexation reactions. In addition, the phenyl-ethyn-phenyl spatial group of **A4,** in the complex [Ir(ppy)_2_(**A4**)]^+^, does not suffer the cleavage of the ether C–O bonds, ensuring stability and electronic conjugation.

### 2.2. Absorption Spectra and Luminescence

The UV–Vis absorption and photoluminescence spectra of [Ir(ppy)_2_(**B1**)]^+^ and [Ir(ppy)_2_(**A4**)]^+^ in acetonitrile solution are depicted in Figure 4 and the photophysical properties are summarized in Table 1. Both complexes show strong absorption bands in the UV region and a very similar photoluminescence at about 500 nm, in deaerated solutions. Density functional theory (DFT) calculations were run to support the assignment of the experimental bands. The simulated absorption spectra are in perfect agreement with the experimental ones. In the case of [Ir(ppy)_2_(**B1**)]^+^, all the main electronic transitions are mixed metal/ligand-to-ligand charge transfer (^1^MLLCT), starting from Ir(III) d orbitals and π orbitals on the phenyl moiety of the two phenylpyridines and ending to π* orbitals on the pyridine moieties. Qualitative evidence can be found in the Electron-Density Difference Map (EDDM) of the first electronic transition, reported as inset in Figure 4a. On the other hand, the absorption spectrum of [Ir(ppy)_2_(**A4**)]^+^ is characterized by a strong absorption at 313 nm, attributable to a pure intra-ligand transition involving the π−π* orbitals of the aromatic spacer (see EDDM of transition 7 in Figure 4b). This is not the only difference with respect to [Ir(ppy)_2_(**B1**)]^+^ indeed, alternating with the typical ^1^MLLCT in the low energy portion of the spectrum, DFT calculations identify a set of ligand-to-ligand charge transfer (^1^LLCT), involving the HOMO that is centered on the aromatic spacer and virtual orbitals centered on the phenyl and/or pyridine rings (see EDDMs of transition 3 in Figure 4b).

Despite these peculiarities in the absorption spectrum, both [Ir(ppy)_2_(**B1**)]^+^ and [Ir(ppy)_2_(**A4**)]^+^ exhibit the same blue luminescence, with almost identical profiles and energies. The fine vibronic progression of the emission bands indicates a strongly ligand-centered nature of the emitting state, while the observed lifetimes are consistent with emission from a triplet state. This is consistent with the substantial decrease of emission in aerated solutions, compared to argon-degassed solutions, due to the quenching effect of oxygen because of triplet-triplet energy transfer. These results can be rationalized with the help of DFT calculations. Indeed, the spin-density distributions for the lowest-lying triplet state of [Ir(ppy)_2_(**B1**)]^+^ and [Ir(ppy)_2_(**A4**)]^+^ display the same topology (Figure 5), involving the metal center and one of the two phenylpyridine ligands. This strongly ligand-centered state is expected to be responsible for the phosphorescence of both complexes. Therefore, the probable mechanism is that after excitation to the singlet manifold of excited states, efficient intersystem crossing to the triplet manifold occurs due to the strong spin-orbit coupling given by the heavy iridium center.

### 2.3. Electrochemistry

The electrochemical behavior of cationic iridium cyclometalated complexes is well known: one electron ligand-based reduction and one electron metal-based oxidation are typically observed for this type of molecules [36,37,38,39].

The cyclic voltammogram (CV) of 1 mM [Ir(ppy)_2_(**B1**)]^+^ acetonitrile solution (0.1 M TBAPF_6_ as supporting electrolyte) at a scan rate of 200 mV/s, shows a reversible Ir(III)/Ir(IV) oxidation process at 0.953 V vs. Fc/Fc^+^ (Table 1). DFT calculations confirm this assignment, indicating a contribution to the HOMO of 38% from the Ir atom (and 30% from the phenyl moiety of each ppy). In the same conditions, the CV of [Ir(ppy)_2_(**A4**)]^+^ shows an irreversible oxidation wave at +0.494 V vs. Fc/Fc^+^, assigned to the oxidation of the aromatic amine of the ligand. Based on DFT calculations, the HOMO is centered exclusively (100%) on the (phenylethynyl)aniline moiety on **A4**, with no contribution from the metal. This feature is suitable for possible glassy carbon functionalization, as previously reported [32,40]. A second irreversible peak at +0.891 V is assigned to the typical Ir(III)/Ir(IV) oxidation.

Cathodic scans of acetonitrile solutions of the two iridium complexes show one electron electrochemically irreversible reductions at potentials more negative than −2 V. These processes can be localized on the organic part of the complexes. In particular, both complexes show common irreversible reduction waves at −2.363 V, assigned to the reduction of the phenylpyridine moieties [39]. The second reduction of [Ir(ppy)_2_(**B1**)]^+^ is set to a more negative potential (−2.595 V) and can be attributed to the ligand **B1**. On the other way, the first reduction process of [Ir(ppy)_2_(**A4**)]^+^ occurs at a less negative potential (−2.093 V) and involves the chelating moiety of **A4**.

## 3. Materials and Methods

All chemicals and solvents were purchased and used without further purification. The reaction monitoring via thin-layer chromatography (TLC) was performed on Fluka silica gel TLC-PET foils GF 254, particle size 25 mm, medium pore diameter 60 Å. Di-2-pyridylketone, 2-phenylpyridine, IrCl_3_·3H_2_O and all other reagents and solvents were of reagent grade and used as received without any further purification. Generous loan of IrCl_3_·3H_2_O from Johnson Matthey is gratefully acknowledged. Acetonitrile was distilled over calcium hydride just before use. All the reactions involving the metal complexes or precursor were routinely performed under a nitrogen atmosphere using standard Schlenk techniques. The chloro-bridged iridium(III) binuclear precursor, [Ir(ppy)_2_Cl]_2_, was synthesized reacting IrCl_3_·3H_2_O and 2-phenylpyridine; following the microwave-accelerated procedure recently reported by Orwat et al. [41]**.**

### 3.1. Synthesis of Ligands

Di(pyridin-2-yl)methanol (**L0**) was prepared as previously reported [10]. Starting from **L0**, three di(pyridin-2-yl)methan-ether intermediates were synthesized as described in the Appendix A. The final ligands, bearing BOC protected (**B1**) or free (**A1**-**A4**) amino functionality, were obtained as described hereafter.

**B1**: tert-butyl 3-(dipyridin-2-ylmethoxy)propylcarbamate

A suspension of sodium hydride (360 mg, 15 mmol, 5 equiv) in dry THF (5 mL) was added to a solution of di(pyridin-2-yl)methanol (553 mg, 3.00 mmol) in dry THF (5 mL) under nitrogen at ambient temperature. After 1 h, when the evolution of hydrogen gas had ceased, the resulting dark blue solution was transferred to a solution of 3-bromopropylamine-BOC (929 mg, 3.91 mmol, 1.3 equiv.) in dry THF (5 mL). The reaction mixture was stirred at ambient temperature for 12 h, and the reaction was terminated by the addition of saturated aqueous NH_4_Cl (8 mL). Extraction with CH_2_Cl_2_ (4 × 40 mL) followed by washing of the organic phase with saturated aqueous NH_4_Cl (5 mL), drying (Na_2_SO_4_), and evaporation of the solvent gave a crude product which was chromatographed (eluent CH_2_Cl_2_-CH_3_OH 98-2) to give 571.8 mg, 1.67 mmol, 56% yield. ^1^H NMR (400 MHz, CDCl_3_) δ (ppm): 8.48 (d, *J* = 5.1 Hz, 2H), 7.59 (m, *J* = 7.7 Hz, 2H), 7.45 (d, *J* = 7.9 Hz, 2H), 7.09 (t, *J* = 6.3 Hz, 2H), 5.53 (s, 1H), 3.55 (t, *J* = 5.8 Hz, 2H), 3.25 (m, 2H), 1.76 (m, 2H), 1.35 (s, 9H). ^13^C NMR (100.44 MHz, CDCl_3_) δ (ppm): 160.0, 156.0, 149.0, 136.3, 122.0, 121.2, 85.4, 78.6, 67.5, 39.6, 29.4, 28.3, 21.0. MS (ESI+) *m/z* calculated for C_19_H_26_N_3_O_3_ ([M + H]^+^): 344.20; found: 344.32.

**A1**: 3-(dipyridin-2-ylmethoxy)propan-1-amine

**B1** (180 mg, 0.52 mmol) was treated with CF_3_COOH (0.160 mL, 2.1 mmol, 4 equiv.) in a solution of dry CH_2_Cl_2_ (3 mL) for 2.5 h. At the solution was added NaOH (1M, 6 mL) followed by extraction with CH_2_Cl_2_ (3 × 2 mL). Evaporation of the solvent gave **A1** as a crude product (53 mg, 0.218 mmol, 42% yield). ^1^H NMR (400 MHz, CDCl_3_) δ (ppm): 8.55 (d, *J* = 5.1 Hz, 1H), 7.68 (t, *J* = 7.7 Hz, 2H), 7.55 (d, *J* = 7.9 Hz, 2H), 7.16 (t, *J* = 6.2 Hz, 2H), 5.59 (s, 1H), 3.63 (t, *J* = 6.2 Hz, 2H), 2.85 (t, *J* = 6.7 Hz, 2H), 1.82 (m, 2H). ^13^C NMR (100.44 MHz, CDCl_3_) δ (ppm): 160.6, 149.4, 136.8, 122.6, 121.5, 86.2, 67.8, 39.8, 33.6. MS (ESI^+^) *m/z* calculated for C_14_H_18_N_3_O ([M + H]^+^): 244.14; found: 244.20.

**A2**: 4-(3-(di(pyridin-2-yl)methoxy)prop-1-en-1-yl)aniline

**A2** was synthesized from an equimolar mixture of 2,2’-(allyloxymethylene)dipyridine (354 mg, 1.56 mmol) and 4-tert-butyl-4-iodophenylcarbamate (500 mg, 1.56 mmol) dissolved into 50 mL of degassed ethanolamine containing catalytic amounts of Pd(OAc)_2_ (15 mg). The mixture was stirred for 12 h at 100 °C, under nitrogen atmosphere. The resulting solution was first treated with CF_3_COOH (0.160 mL, 2.1 mmol, 4 equiv.) in a solution of dry CH_2_Cl_2_ (3 mL) for 2.5 h, and then extracted with dichloromethane. The organic solution was dried under vacuum. The crude product was chromatographed (eluent CH_2_Cl_2_-CH_3_OH 98-2) to give 76 mg, 0.182 mmol, 12% yield. ^1^H NMR (400 MHz, CDCl_3_) δ (ppm): 8.51 (d, *J* = 5.5 Hz, 2H), 7.84 (t, *J* = 7.5 Hz, 2H), 7.64 (t, *J* = 7.6 Hz, 2H), 7.13 (t, *J* = 6.0 Hz, 2H), 7.02 (d, *J* = 8.2 Hz, 2H), 6.53 (d, *J* = 8.2 Hz, 2H), 6.48 (s, 1H), 6.29 (d, *J* = 13.6 Hz, 1H), 5.90–5.83 (m, 1H), 3.27 (d, *J* = 7.5 Hz, 2H). ^13^C NMR (100.44 MHz, CDCl_3_) δ (ppm): 163.8, 157.5, 147.8, 145.6, 136.9, 135.1, 133.0, 127.4, 122.2, 121.9, 121.2, 115.1, 55.6, 45.6. MS (ESI+) *m/z* calculated for C_20_H_20_N_3_O ([M + H]^+^): 317.39; found: 317.38.

**A3**: 4-(3-(benzhydryloxy)prop-1-ynyl)aniline

**A3** was synthesized from a mixture of (2,2’-((prop-2-ynyloxy)methylene)dipyridine) (516 mg, 2.30 mmol) and 4-iodoaniline (580 mg, 2.65 mmol, 1.1 equiv.) dissolved into 30 mL of diethylamine containing catalytic amounts of PdCl_2_(PPh_3_)_2_ (15 mg) and CuI (7 mg). The mixture was stirred for 3 h at room temperature, under nitrogen atmosphere. The solution was first hydrolyzed with distilled water, and then extracted with dichloromethane (4 × 5 mL). The organic solution was dried under vacuum. The crude product was chromatographed on aluminum oxide (eluent CH_2_Cl_2_-CH_3_OH 99-1) to give 62 mg, 0.20 mmol, 9% yield. ^1^H NMR (400 MHz, CDCl_3_) δ (ppm): 8.54 (d, *J* = 4.6 Hz, 2H), 7.70–7.62 (m, 4H), 7.23–7.13 (m, 4H), 6.53 (d, *J* = 8.8 Hz, 2H), 5.94 (s, 1H), 4.47 (s, 2H), 3.78 (s, 2H). ^13^C NMR (100.44 MHz, CDCl_3_) δ (ppm): 159.9, 149.5, 147.1, 136.8, 133.3, 122.7, 122.0, 114.7, 112.9, 87.7, 84.0, 82.4, 57.7. MS (ESI+) *m/z* calculated for C_20_H_18_N_3_O ([M + H]^+^): 316.14; found: 316.37.

**A4**: (4-((4-((dipyridin-2-ylmethoxy)methyl) phenyl)-ethynyl)aniline

**A4** was synthesized from an equimolar mixture of 2,2’-((4-iodobenzyloxy)methylene)dipyridine (731 mg, 1.82 mmol) and 4-ethynylaniline (213 mg, 1.82 mmol) dissolved into 50 mL of diethylamine containing catalytic amounts of PdCl_2_(PPh_3_)_2_ (7 mg) and CuI (5 mg). The mixture was stirred for 3 h at room temperature, under nitrogen atmosphere. The solution was first hydrolyzed with distilled water, and then extracted with dichloromethane. The organic solution was dried under vacuum. The crude product was chromatographed (eluent CH_2_Cl_2_-CH_3_OH 95-5) to give 370 mg, 1.86 mmol, 52% yield. ^1^H NMR (400 MHz, CD_2_Cl_2_) δ (ppm): 8.52 (s, 2H), 7.74 (t, *J* = 7.3 Hz, 2H), 7.60 (d, *J* = 7.9 Hz, 2H), 7.47 (d, *J* = 7.9 Hz, 2H), 7.35 (d, *J* = 8.6 Hz, 2H), 7.32 (d, *J* = 8.5 Hz, 2H), 7.21 (s, 2H), 6.65 (d, *J* = 9.4 Hz, 2H), 5.68 (s, 1H), 5.37 (s, 2H), 4.61 (s, 2H). ^13^C NMR (100.44 MHz, CD_2_Cl_2_) δ (ppm): 160.6, 149.1, 147.2, 138.0, 136.6, 132.8, 131.3, 128.0, 123.4, 122.6, 114.7, 112.0, 90.2, 87.2, 85.5, 70.9. MS (ESI+) *m/z* calculated for C_26_H_22_N_3_O ([M + H]^+^): 392.18; found: 392.40.

### 3.2. Synthesis of Iridium Complexes

[Ir(ppy)_2_(**B1**)]^+^

A suspension of [Ir(ppy)_2_Cl]_2_ (200 mg, 0.184 mmol) and **B1** (126 mg, 0.367 mmol, 2 equiv.) in CH_2_Cl_2_ and CH_3_OH (20 mL, 1:1 *v*/*v*) was heated to reflux and stirred under inert atmosphere for 2 h. The resulting yellow solution was cooled to room temperature. The solution volume was then reduced to approximately 10 mL. After filtration and evaporation the crude product was chromatographed on aluminum oxide (eluent CH_2_Cl_2_-CH_3_OH 96-4) to give 30 mg, 0.034 mmol, 9% yield of [Ir(ppy)_2_(**B1**)]Cl. ^1^H NMR (400 MHz, CDCl_3_) δ (ppm): 8.32 (d, *J* = 5.47 Hz, 1H), 8.12–7.94 (m, 6H), 7.90–7.85 (m, 3H), 7.75 (d, *J* = 7.86 Hz, 1 H), 7.56–7.51 (m, 3H), 7.02–6.87 (m, 7H), 6.81 (t, *J* = 7.35 Hz, 1H), 6.19 (d, *J* = 7.52 Hz, 2H), 6.15 (d, *J* = 7.52 Hz, 2H), 5.90 (s, 1H), 5.74 (t, *J* = 5.47 Hz, 1H), 3.74–3.64 (m, 4H), 3.34 (m, 2H), 1.95 (m, 2H), 1.39 (s, 9H). ^13^C NMR (100.44 MHz, CDCl_3_) δ (ppm): 158.0, 152.8, 152.0, 151.3, 149.0, 147.0, 144.0, 143.0, 140.2, 139.9, 139.0, 132.2, 131.9, 131.0, 130.9, 130.8, 130.5, 130.0, 128.0, 126.4, 125.9, 125.6, 125.4, 123.8, 124.6, 124.4, 124.0, 123.3, 123.0, 120.2, 81.6, 79.3, 78.9, 77.1, 69.9, 37.7, 31.1, 28.5, 28.1. MS (ESI+) m/z calculated for IrC_41_H_41_N_5_O_3_ ([M]^+^): 844.28; found: 844.10.

[Ir(ppy)_2_(**A4**)]^+^

A suspension of [Ir(ppy)_2_Cl]_2_ (92 mg, 0.086 mmol) and **A4** (70 mg, 0.178 mmol) in CH_2_Cl_2_ and CH_3_OH (20 mL, 1:1 *v*/*v*) was heated to reflux and stirred under inert atmosphere for 2 h. The resulting yellow solution was cooled to room temperature. The solution was filtered and the solid eliminated. The solution volume was then reduced to approximately 10 mL. Precipitation of the final complex was achieved by adding to the concentrated solution a 5-fold excess of ammonium hexafluorophosphate dissolved in H_2_O (2 mL) and by stirring the mixture for 15 min. After filtration, the precipitate was washed with diethyl ether and the crude product was chromatographed (eluent CH_2_Cl_2_-CH_3_OH 98-2) to give 29 mg, 0.028 mmol, 16% yield of [Ir(ppy)_2_(**A4**)]PF_6_. ^1^H NMR (400 MHz, CD_2_Cl_2_) δ (ppm): 8.45 (d, *J* = 4.80 Hz, 1H), 8.37 (d, *J* = 4.70 Hz, 1H), 8.21 (d, *J* = 6.88 Hz, 1H), 8.11 (d, *J* = 4.25 Hz, 1 H), 8.09 (d, *J* = 4.30 Hz, 1H), 7.35 (d, *J* = 8.60 Hz, 2H), 8.05–7.99 (m, 2H), 7.89–7.81 (m, 3H), 7.75 (t, *J* = 7.91 Hz, 1H), 7.60–7.56 (m, 2H), 7.46 (d, *J* = 8.20 Hz, 2H), 7.35–7.31 (m, 3H), 7.22 (d, *J* = 8.20 Hz, 2H), 7.07–7.00 (m, 2H), 6.96 (t, *J* = 6.92 Hz, 1H), 6.90 (t, *J* = 6.90 Hz, 1H), 6.84 (t, *J* = 7.32 Hz, 1H), 6.79 (t, *J* = 6.80 Hz, 1H), 6.76 (d, *J* = 5.60 Hz, 1H), 6.66 (d, *J* = 8.20 Hz, 2H), 6.21 (d, *J* = 6.88 Hz, 1H), 6.13 (d, *J* = 6.88 Hz, 1H), 5.92 (s, 1H), 4.87 (d, *J* = 13.00 Hz, 1H), 4.75 (d, *J* = 13.00 Hz, 1H), 4.00 (s, 2H). ^13^C NMR (100.44 MHz, CD_2_Cl_2_) δ (ppm): 177.4, 168.9, 167.4, 159.9, 157.9, 152.9, 152.5, 151.8, 150.9, 150.4, 148.5, 147.5, 146.5, 143.6, 139.6, 138.9, 135.7, 133.2, 132.8, 132.5, 132.1, 131.8, 131.5, 127.2, 127.0, 126.3, 125.6, 125.25, 123.3, 123.0, 122.8, 120.7, 115.0, 114.2, 111.7, 91.2, 86.6, 80.5, 72.2. MS (ESI+) *m/z* calculated for IrC_48_H_37_N_5_O ([M]^+^): 892.26; found: 892.24.

### 3.3. Characterization

^1^H and ^13^C NMR spectra were recorded on a JEOL ECP 400 FT-NMR spectrometer (^1^H NMR operating frequency 400 MHz) (Jeol, Tokyo, Japan). Chemical shifts are reported relative to TMS (δ = 0) and referenced against solvent residual peaks (acetone-d6 and DMSO-d6).

UV–Vis absorption spectra were recorded with a double-beam Perkin-Elmer Lambda 20 UV–Vis spectrophotometer (Perkin Elmer, Waltham, MA, USA). Photoemission spectra, luminescence lifetimes and quantum yields were acquired with a HORIBA Jobin Yvon IBH Fluorolog-TCSPC spectrofluorometer (Horiba, Kyoto, Japan). Fluorescence quantum yields Φ were determined using quinine bisulfate (0.1 N H_2_SO_4_) as standard (Φ = 0.546). Refractive index corrections were made to adjust for different solvents used. Luminescence lifetimes were determined by time-correlated single-photon counting. Excitation with nanosecond pulses of 370 nm light generated by a NanoLED pulsed diode was used. The emission data were collected using a spectral bandwidth of 2–10 nm. The data were collected into 2048 channels to 10,000 counts in the peak channel. Emission decay data were analyzed using the software DAS6 (TCSPC Decay Analysis Software, Horiba, Kyoto, Japan). Mass spectra were recorded using an XCT PLUS electrospray ionization–ion trap (ESI–IT) mass spectrometer (Agilent Italy, Milan). Samples were dissolved in methanol/water (9:1) solution with formic acid (0.1%). In the spectra description the abbreviation [M]^+^ was used for the molecular ion. The reported values are in atomic mass units. The m/z scan range was 50–2000. Electrochemistry was carried out in acetonitrile with tetrabutylammonium hexafluorophosphate (TBAPF_6_) 0.1 M as supporting electrolyte, using a standard three electrode cell configuration (glassy carbon working electrode, Pt counter electrode, 3 M KCl Calomel reference electrode) and Autolab PGSTAT302N (Metrohm Autolab, Herisau, Switzerland) electrochemical analyzer controlled by a PC. Acetonitrile was distilled over calcium hydride just before use, and TBAPF_6_ was obtained as metathesis reaction between KPF_6_ and tetrabutylammonium iodide, recrystallized three times from 95% ethanol, and dried in a vacuum oven at 110 °C overnight. Positive feedback Ir compensation was applied routinely. All measurements were carried out under Ar in anhydrous deoxygenated solvents. Ferrocene (Fc) was used as an internal standard, and half-wave potentials are reported against the Fc(0/+1) redox couple (measured E_1/2_(0/+1) 0.386 V).

### 3.4. Computational Details

All calculations were performed with the Gaussian 16 program package [42], employing the Density Functional Theory (DFT) and its Time-Dependent extension TD-DFT [43,44], The hybrid exchange-correlation functional PBE1PBE [45] was employed together with the 6-31G** basis set [46] and the LanL2DZ basis set and effective core potential (Ir atom) [47,48]. The solvent effect was included using the polarizable continuum model (CPCM method) [49,50], with acetonitrile as solvent. Geometry optimizations were carried out for ground states, without any symmetry constraints. The nature of all stationary points was verified via harmonic vibrational frequency calculations, confirming that every stationary point found by geometry optimizations was actually a minimum on the corresponding potential-energy surface (no imaginary frequencies were found). Electronic transitions were computed from the ground state, as vertical excitation with linear response solvation by TD-DFT, employing the ground state optimized geometries. A total of 128 singlet excited states was computed for each compound, the electronic distribution and the localization of the singlet excited states were visualized using Electron-Density Difference Maps. GaussSum 2.2.5 [51] was used to simulate the theoretical UV–Vis spectra and for EDDMs calculations [52,53]. Molecular-graphic images were produced using the UCSF Chimera package from the Resource for Biocomputing, Visualization, and Informatics at the University of California, San Francisco [54].

## 4. Conclusions

Five new ligands with amino or carbamates functional groups, were synthesized and characterized. Two new iridium complexes were obtained and their redox behavior and optical properties reported. The new species exhibit strong absorption in the UV region, due to mixed metal/ligand-to-ligand charge transfers and relatively strong and long-lived luminescence (10^−6^ s) compatible with triplet excited states. Several redox processes can be demonstrated in each complex and assigned to specific components.

In the case of **A1**, the stronger nucleophilicity of the primary amino group with respect the aromatic ones, resulted in several side and decomposition products when reacted with [Ir(ppy)_2_Cl]_2_. Therefore, the synthetic procedure must proceed with the BOC protection (**B1**), to preserve the terminal NH_2_ moiety. Nevertheless, the following deprotection of the amino group of [Ir(ppy)_2_(**B1**)]^+^ leads to de-complexation of the di(pyridin-2-yl) ligand moiety. Therefore, to avoid any interference of the terminal amino group during the complexation, we introduced different spacers on the di(pyridin-2-yl) chelating moiety, obtaining other three new ligands (**A2**–**A4**). Among these ligands, those containing alkene or alkyne units vicinal to the oxy-methylene portion (**A2** and **A3**), are instable in the complexation conditions, suffering the cleavage of the ether C–O bonds. At the contrary, **A4** and **B1** were successfully reacted with [Ir(ppy)_2_Cl]_2_, obtaining two new stable and luminescent iridium complexes, namely [Ir(ppy)_2_(**A4**)]^+^ and [Ir(ppy)_2_(**B1**)]^+^.

The modification of the spacer does not affect in a substantial way the photophysical properties of the studied products, since the modification is far from the metal core. Conversely, the introduction of a free amino group greatly affects the redox properties of the products causing the appearance of irreversible oxidation peak at low potential. This oxidation can be exploited for electrochemical surface functionalization, enabling the approach of supporting intact organometallic molecules as emitters or catalysts on solid surfaces [55,56].

## Data Availability

The datasets generated during the current study are available from the corresponding author on reasonable request.

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
