# Peer review of "Dipyridylmethane Ethers as Ligands for Luminescent Ir Complexes"

_molecules, 2021, doi:10.3390/molecules26237161_

Round 1

Reviewer 1 Report

The structure of the article is not appropriate. Chapter 2-Results should be integrated in Chapter 4- Materials and Methods.

Isn't it very clear what the purpose of this article is.

The statement ”The obtained ether-based intermediates were modified through cross coupling reactions to extend the π system by insertion of highly conjugated units such as phenyl, allyl, ethynyl.” is incorrect, as newly introduced conjugate systems will not be able to conjugate with the bi-pyridyl system due to the ether group (as seen in Figure 4).

It is not very clear the choice of the 2 classes of ligands (B and A). Wouldn't it have been better for the two classes of ligands to be similar (respectively with the free amino group and the protected amino group)?

In order to have the possibility of comparison, the option of obtaining [Ir(ppy)2(A1)]+ by hydrolysis of  [Ir(ppy)2 (B1)]+ could be tried. Also if [Ir(ppy)2(A3)]+ is stable then it is necessary to obtain and the variant with protected amino group.

Much more pertinent explanations should be given regarding the rupture of the C-O bond in the case of some ligands.

The list of bibliographical references does not respect any of the styles agreed (ACS or Chicago style) by the journal.

Other comments are in the attached document.

Reviewer 2 Report

In general, the presented paper concerning luminescent Ir complexes, their synthesis and spectral properties is a good work. However the paper needs to be improved.

Most important is that the layout of this paper is not clear. In the part ‘2. Results’, particularly ‘2.1. Synthesis of ligands’ there are a complete preparation protocols which belongs to the experimental part but not for results. In this part some comments and remarks for synthetic process are recommended along with Schemes of reactions with conditions (not only compounds, see Figures 1, 2 and 3) to visualize synthetic route. In means that the lines from 74 to 118; 123 to 156, and 160 to 196 must be shifted to part ‘4.1. Synthesis’. In addition, the preparation of [Ir(ppy)2Cl]2, which is not common substance, is also missing and must be added. In line 320 authors mentioned IrCl3·3H2O, but no word about further usage.

Finally, in my opinion the part ‘2. Results’ and ‘3. Discussion’, e.g, part ‘2.1. Synthesis of ligands’ with ‘3.1. Syntheses’ should be merged, which better express the scientific value of this investigations and will be more clear. After those changes this article will be more friendly for readers and better understandable.

Additionally, inline 66 – this is not scheme – this is figure.

Reviewer 3 Report

The author's work has very high reference value for other researchers. However, the expression of the conclusion is not clear enough,the part relating to references 33, 34, 53-54 would be more appropriate in the discussion section.

Round 2

Reviewer 1 Report

The changes made have improved the paper. It would have been useful to upload the article in the final form, not only in the form of a draft with modifications. 

Some changes are needed. First of all, the statement " This observation led us to consider different spacers in order to increase the distance and the electronic conjugation between the ligand portion and the metal centre (A2) page 4 -rows 108,109" is not correct. The increase in the distance between the amino group and the ligand has no influence on the conjugation due to the presence of the ether group that prevents conjugation, even if there are unsaturated functional groups on the spacer. This is also evident from DFT calculations or absorbance and luminescence spectra.

Considering that one of the main objectives of the article is the exploration of new different synthetic pathways to obtain ligands with functionalized spacer, I think that a reaction scheme that would include the S1-S4 schemes from the Suplementary material would be useful in the paper.

I think the statement ”This is a very promising research field where we demonstrated that catalytic Turn Over Numbers can be increased by four order of magnitude with respect the homogeneous counterpart [56,57]” is a bit forced in the Conclusions and it doesn't make much sense.

Author Response

Reviewer #1 Comments and Suggestions for Authors

The changes made have improved the paper. It would have been useful to upload the article in the final form, not only in the form of a draft with modifications.

Answer: Reviewer #1 is totally right; we apologise for the reading difficulty caused by our lack.

Some changes are needed. First of all, the statement "This observation led us to consider different spacers in order to increase the distance and the electronic conjugation between the ligand portion and the metal centre (A2) page 4 -rows 108,109" is not correct. The increase in the distance between the amino group and the ligand has no influence on the conjugation due to the presence of the ether group that prevents conjugation, even if there are unsaturated functional groups on the spacer. This is also evident from DFT calculations or absorbance and luminescence spectra.

Answer: based on what suggested by Reviewer #1, the sentence has been modified as follow: “This observation led us to consider different spacers in order to increase the rigidity and the distance between the ether junction and the terminal amino group (A2–A4).”

Considering that one of the main objectives of the article is the exploration of new different synthetic pathways to obtain ligands with functionalized spacer, I think that a reaction scheme that would include the S1-S4 schemes from the Suplementary material would be useful in the paper.

Answer: as suggested by Reviewer #1, we integrated the aforementioned schemes S1-S4 into the manuscript (new Scheme 2).

I think the statement ”This is a very promising research field where we demonstrated that catalytic Turn Over Numbers can be increased by four order of magnitude with respect the homogeneous counterpart [56,57]” is a bit forced in the Conclusions and it doesn't make much sense.

Answer: considering Reviewer #1's comment, the sentence has been removed from the Conclusions.

Reviewer 2 Report

No comments

Author Response

We thank the referee 2, all requests were resolved in round 1. There are no problems to be solved in round 2
